# Controlling Molecular Orientation of Small Molecular Dopant-Free Hole-Transport Materials: Toward Efficient and Stable Perovskite Solar Cells

**DOI:** 10.3390/molecules28073076

**Published:** 2023-03-30

**Authors:** Wenhui Li, Chuanli Wu, Xiuxun Han

**Affiliations:** Institute of Optoelectronic Materials and Devices, Faculty of Materials Metallurgy and Chemistry, Jiangxi University of Science and Technology, Ganzhou 341000, China; liwenhui@jxust.edu.cn (W.L.);

**Keywords:** perovskite solar cells, dopant-free hole-transport materials, efficiency, stability, molecular orientation

## Abstract

Perovskite solar cells (PSCs) have great potential for future application. However, the commercialization of PSCs is limited by the prohibitively expensive and doped hole-transport materials (HTMs). In this regard, small molecular dopant-free HTMs are promising alternatives because of their low cost and high efficiency. However, these HTMs still have a lot of space for making further progress in both efficiency and stability. This review firstly provides outlining analyses about the important roles of molecular orientation when further enhancements in device efficiency and stability are concerned. Then, currently studied strategies to control molecular orientation in small molecular HTMs are presented. Finally, we propose an outlook aiming to obtain optimized molecular orientation in a cost-effective way.

## 1. Introduction

Solar photovoltaic (PV) technology, which converts sunlight directly to electricity in a green, safe and efficient way, plays a central role in fulfilling the net-zero targets (for greenhouse gas emission, especially CO_2_) set for around 2050 [1]. In the current PV market, crystalline silicon (c-Si) solar cells are in a dominant position. Despite an 88% decline in their levelized cost of energy (LCOE) between 2010 and 2021 [2], fossil fuels still accounted for a global share of 62% in electricity generation in 2021 [3]. Hence, continuous efforts to reduce the LCOE of solar PV are needed to enhance their cost competitiveness as compared to fossil fuels.

Perovskite solar cells (PSCs) are a new type of PV technology developed in recent decades in which the absorber material is composed of metal halide perovskites with a general chemical formula of ABX_3_. Generally, the A site is a monovalent cation, such as [HC(NH_2_)_2_]^+^ (FA^+^), CH_3_NH_3_^+^ (MA^+^) or Cs^+^. The B site is mainly Pb^2+^, and the X site is a halide ion (I^−^, Br^−^ or Cl^−^). To fabricate a PSC, the perovskite layer is usually sandwiched between an electron-transport layer (ETL) and a hole-transport layer (HTL), constituting a n–i–p device (Figure 1a) or p–i–n device (Figure 1b), which depends on the relative position of the charge carrier transport layers (CTLs). Upon illumination, the photogenerated electrons and holes in the perovskite layer will be selectively transported to the corresponding electrode through CTLs, and the charge carrier recombination at the two perovskite interfaces can be mitigated. Therefore, CTLs are indispensable for efficient PSCs despite the fact that metal halide perovskites possess the property of ambipolar charge transport. Practically, the CTL, especially the HTL, plays a key role in the development of PSCs. In 2009, Miyasaka et al. first introduced MAPbI_3_ and MAPbBr_3_ nanocrystals to sensitize a meso-TiO_2_ electrode in dye-sensitized solar cells (DSCs), obtaining a power conversion efficiency (PCE) of 3.81% and 3.13%, respectively [4]. However, perovskite nanocrystals degraded very fast in the liquid electrolyte. In 2012, Park et al. used solid spiro-OMeTAD (2,2′,7,7′-tetrakis[N,N-di(4-methoxyphenyl)amino]-9,9′-spirobifluorene) to replace the liquid electrolyte and achieved a solid DSC with PCE of 9.7% [5]. Thereafter, PSCs have embarked on a fast track of efficiency development [6,7,8]. Very recently, a certified PCE of 25.7% and 32.5% has been reported for single-junction PSCs and perovskite-silicon tandem solar cells [9,10], respectively, demonstrating perovskite PV as an efficient solar PV technology.

In addition to PCE, manufacturing cost and long-term stability are two other key factors determining the LCOE of a certain photovoltaic technology [11]. While the manufacture of PSCs is compatible with low-temperature (<300 °C) and high-volume processing technologies [12], its manufacturing cost strongly depends on the materials [13]. For modules containing expensive HTL and metal electrodes, the manufacturing cost could be significantly higher than that of the c-Si PV [11]. In recent PSCs with recorded PCEs, they all use spiro-OMeTAD or PTAA (poly[bis(4-phenyl)(2,4,6-trimethylphenyl)amine]) as the HTL [14,15,16]. However, the synthesis and purification procedures of spiro-OMeTAD and PTAA are tedious, making them prohibitively expensive for future commercialization [17].

Moreover, both spiro-OMeTAD and PTAA generally need p-doping due to their low intrinsic hole mobility (~10^−5^ cm^2^ V^−1^ s^−1^) [18]. Upon doping, the oxidized HTM molecules will increase the charge carrier density and thereby the conductivity, whereas it will induce a series of stability issues [18,19]. For example, it was found that the oxidized spiro-OMeTAD molecules will degrade or be reduced under thermal stress [20,21]. In addition, the Lewis acid p-dopant in the HTL may attack the perovskite surface, resulting in rapid performance degradation [22]. Therefore, the usage of p-doped HTL limits the lifetime of the PSCs. Studies on cost analysis have shown that the LCOE of perovskite PV strongly depends on the lifetime [23,24]. Meng et al. estimated the dependence of LCOE on lifetime by assuming a module efficiency of 19% [25]. A LCOE of ~0.15 USD kW^−1^ h^−1^ was calculated for the module with a lifetime of 5 years, which is much higher than the c-Si PV.

Hence, the development of low-cost and dopant-free hole-transport materials (HTMs) is of significant importance for the commercialization of perovskite PV, and it has become a research hotspot. Generally, dopant-free HTMs can be classified into three categories, including polymers, small molecules, and inorganic compounds. Among them, polymeric HTMs possess the highest PCE for dopant-free HTMs [16]. However, their large-scale application is limited by their extremely high cost [26]. Inorganic HTMs seem to be ideal candidates for dopant-free HTMs because of advantages such as high hole mobility, good stability, and low cost, while the efficiency of inorganic HTMs is still lagging behind that of their organic counterparts [27]. Furthermore, recent studies indicate that the interaction between inorganic HTMs and perovskites may accelerate the aging of PSCs [22,28]. The above limitations of polymeric and inorganic HTMs have paved the way for the development of small molecular HTMs [29]. Recently, dopant-free small molecular HTMs have achieved impressive progress. In p-i-n PSCs, a PCE of 24.34% was achieved by a small molecule coded as BTP1 [30], approaching the state-of-art PTAA-based device (25.0%) [16]. More importantly, the synthetic cost of BTP1 is significantly lower than that of PTAA (8.98 vs.~2700 USD/g). In the n-i-p device, a small molecule, BDT-DPA-F, achieved a PCE of 23.12% [31]. The efficiency was slightly lower than the highest record (24.6%) achieved by P3HT (poly(3-hexylthiophene)) [32], but the BDT-DPA-F-based device exhibited ultralong stabilities under both operation and thermal aging conditions. In addition, a phthalocyanine derivate coded as SMe-TPA-CuPc obtained a similar PCE of 23.0% and retained 96% of the initial PCE after being thermally stressed at 85 °C for 3624 h [33]. In this respect, small molecular HTMs have demonstrated their great potential as efficient, low-cost, and stable HTMs.

**Figure 2 molecules-28-03076-f002:**
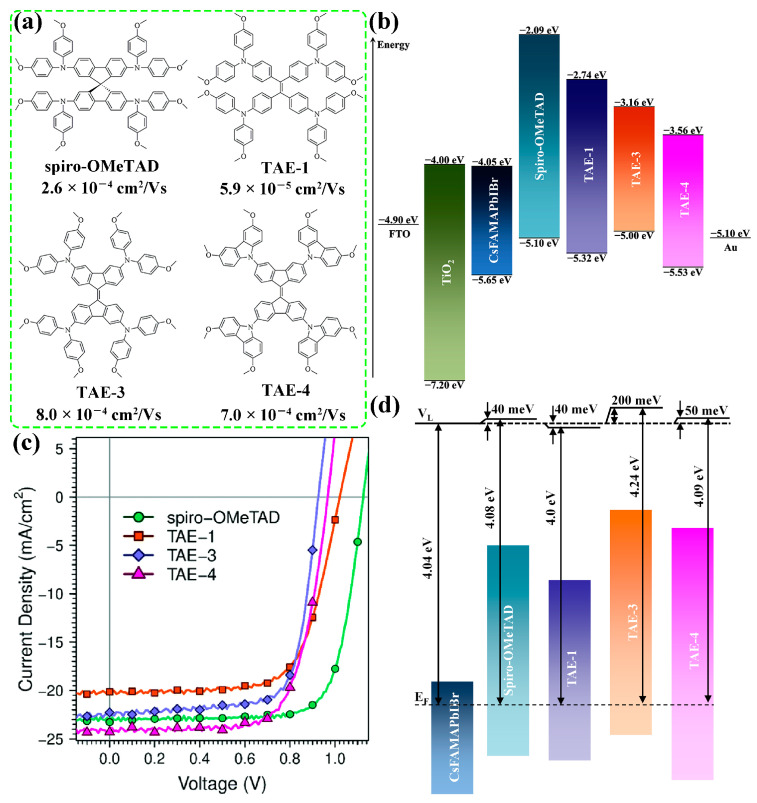
Molecular structures and hole mobilities (**a**), energy levels obtained from CV measurements (**b**), and current-voltage (*J*-*V*) curves (**c**) of spiro-OMeTAD and TAE HTMs. (**d**) V_L_ positions of different perovskite/HTM stacks. Adapted with permission from Ref. [34]. Copyright 2019 The Royal Society of Chemistry.

Despite the encouraging progress of small molecular HTMs, their efficiency is still lagging behind the state-of-art doped HTMs. After numerous efforts devoted to the optimization of molecular design, there are dozens of new HTMs with higher hole mobility than the doped spiro-OMeTAD in efficient PSCs (10^−4^~10^−3^ cm^2^ V^−1^ s^−1^) [35]. However, dopant-free HTMs, even for those with suitable energy levels as obtained from cyclic voltammetry (CV) in solution, often suffer from inferior open-circuit voltage (*V*_OC_) [36,37,38]. Generally, the *V*_OC_ of PSCs is determined by the energy level alignment and the charge carrier recombination [39]. For small molecular HTMs, their lowest unoccupied molecular orbital (LUMO) energy levels are generally lower than that of spiro-OMeTAD, resulting in less sufficient electron blocking at the interface and thereby more charge carrier recombination [38,40,41]. However, the charge carrier recombination cannot explain the difference in *V*_OC_ alone as demonstrated by Gelmetti et al. [34]. In their work, three TAE coded HTMs (Figure 2a) were compared with spiro-OMeTAD. The hole mobilities of three TAE HTMs are not quite distinct from that of spiro-OMeTAD. Furthermore, according to the result from CV measurements, both TAE-1 and TAE-4 exhibited more suitable highest-occupied molecular orbital (HOMO) energy levels than spiro-OMeTAD (Figure 2b), i.e., larger *V*_OC_ would be expected for the former two. However, all three TAE HTMs showed much lower *V*_OC_ in comparison to spiro-OMeTAD (Figure 2c), and the difference in average *V*_OC_ can be up to 170 mV for TAE-4. The comprehensive analysis on charge carrier recombination demonstrated that the largely different *V*_OC_ could not be ascribed to the recombination. The contact potential difference (CPD) measured by Kelvin probe force microscopy (KPFM) revealed that distinct HTMs deposited on perovskite resulted in different shifts in the vacuum level (V_L_) (Figure 2d). Especially, the V_L_ was upward shifted to as large as ~200 meV upon the deposition of TAE-3, implying a same shift in HOMO of TAE-3 compared to the valence band (VB) of perovskite. This result demonstrates that the energy level alignment can be quite different from that predicted by CV measurements. For organic semiconductors, molecular orientation, which is a feature of molecular stacking in the film, plays a determinant role in the interface energy level alignment [42]. Furthermore, molecular orientation of organic semiconductors has significant influence on hole mobility, demonstrated in the field of organic field-effect transistors (OFET) [43,44].

In a previous review focusing on dopant-free small molecular HTMs, molecular orientation was briefly discussed from the view point of molecular structure [45]. However, a certain molecular orientation is the result of synergy effects between the molecular structure and external factors such as substrates and processing methods [46]. In this review, we will first elucidate the role of molecular orientation in determining device physics and stability. Then, the current knowledge on factors influencing molecular orientation will be presented, including molecular structure, thermal treatment and substrates. Lastly, we propose an outlook for the control of molecular orientation.

## 2. Molecular Orientation–Device Performance Relationship

### 2.1. Charge Carrier Transport and Transfer

In organic semiconductors, charge carrier transport is realized via the intramolecular and intermolecular delocalization of π electrons [47]. In polymeric organic semiconductors, long-range intramolecular delocalization is possible due to their elongated π-backbone. Differently, charge carrier transport in small molecular organic semiconductors is more dependent on the intermolecular delocalization. In this respect, charge carrier transport in organic solids depends on the packing details of molecules, which determine the intermolecular π-π overlap. For example, lamellar packing exhibits larger π-π overlap than herringbone packing [47]. In the lamellar packing motif, the π-π overlap degree varies upon the slip displacement and slip angle, which depends on the molecular structure [48,49]. Although charge carrier transport depends on the packing details of molecules, the most efficient charge carrier transport channel is in the direction parallel to the π-π stacking orientation [50,51].

Due to the anisotropy of charge carrier transport, molecular orientation is an important factor that should be taken into consideration when organic semiconductors are applied in electronics or optoelectronics [52]. Upon being deposited onto the substrate, molecules may adopt face-on or edge-on stacking with respect to the substrate. In the face-on configuration, as illustrated in Figure 3, the π-backbone stacks in a manner parallel to the substrate surface. In contrast, π-backbones in edge-on configurations stand on the substrate. Hence, the face-on orientation is preferred in solar cells and OLEDs, while edge-on is the optimal molecular orientation for OFETs due to the in-plane distribution of drain and source electrodes. The mismatch between molecular orientation and current flow direction of the device may result in significantly retarded charge carrier transport. Noh et al. demonstrated that an edge-on oriented porphyrin film exhibits 100 times higher mobility than the face-on oriented one in OFET [53]. In PSCs, several times higher hole mobility was observed in the HTM film with dominant face-on orientation compared to the ones with edge-on orientation [33,54].

In addition to the charge carrier mobility, charge carrier transfer dynamics at the interface also strongly depend on molecular orientation. Similar to the intramolecular charge transport between the electron donor part and the electron acceptor part [55], the interfacial charge transfer depends on the effective electronic coupling, *V_eff_*, at the interface [56]. For the charge transfer between diabatic states *i* and *f*, the relationship between transfer rate *k_i_*_→*f*_ and *V_eff_* is described as $k_{i\to f}=\frac{2\pi }{ћ}V_{eff}^{2}\left(i\to f\right)\rho \left(E_{f}\right)$, where $\rho \left(E_{f}\right)$ is the density of states [56]. Upon depositing a molecule onto a substrate, the interfacial *V_eff_* depends on molecular orientation. Taking the sexithiophene (6T) molecule on Au as an example, density functional theory (DFT) calculations demonstrate that both HOMO and the lowest unoccupied molecular orbital (LUMO) of face-on oriented 6T to delocalize into the Au substrate, while it was not observed in the edge-on system [57]. The faster charge carrier transfer at the interface then can be obtained in the face-on system due to the strong electronic coupling [57,58,59,60]. In 2020, Igci et al. reported three small molecular HTMs coded as CI-B1, CI-B2 and CI-B3, whose conductivities were 7.98 × 10^−8^ S cm^−1^, 5.53 × 10^−7^ S cm^−1^, and 9.60 × 10^−7^ S cm^−1^, respectively [61]. The lowest conductivity of CI-B1 is expected to be not unfavorable for hole extraction [62]. However, both steady-state photoluminescence (ssPL) and time-resolved photoluminescence (TRPL) measurements demonstrated that the CI-B1 HTM possessed the fastest hole extraction due to its face-on orientation. 

### 2.2. Interfacial Energy Level Alignment

In PSCs, the energy level alignment at the perovskite/CTL interfaces plays an essential role in determining the final device performance. Firstly, the energy offset should be large enough for efficient charge carrier extraction. Taking the perovskite/HTL interface as an example, the valence band maxima (VBM) of the perovskite layer should be lower than the HOMO energy level of HTL. Otherwise, a hole extraction energy barrier will form at the interface, resulting in charge carrier recombination and thereby inferior device performance [63]. Westbrook et al. suggested that the energy offset of ~0.07 eV is enough for efficient hole transfer at the perovskite/HLT interface [64]. After the minimum offset being guaranteed, the lower the HOMO of HTL, the larger the *V*_OC_ that can be achieved [65,66].

For a certain organic molecule, the transformation in molecular orientation can induce sizable variations in HOMO levels [67], and the energy level alignment at its interface will change accordingly. In PSCs, the impact of molecular orientation on energy level was impressively revealed by Zhou et al. [54]. The ultraviolet photoelectron spectroscopy (UPS) measurements revealed that the HOMO level of HTL changed from −5.03 to −5.42 eV for the film with 0% and 59.6% face-on orientation, respectively. The similar downward shift of HOMO level was also observed for the a phthalocyanine HTM when molecular orientation was transformed from edge-on to face-on [33]. The dependence of HOMO level on molecular orientation can be rationalized by the change in surface dipole [68]. For a face-on oriented molecule, its π-electron cloud will be exposed to the vacuum with the positively charged backbone lying below, i.e., an intramolecular dipole pointing toward the substrate is formed. Consequently, the position of HOMO level will be lower than that of the molecule in gas phase. In contrast, the direction of the surface dipole will be reversed for edge-on-oriented molecules if it is the H atom exposed to the surface [42].

### 2.3. Device Stability

Due to the removal of dopants, PSCs based on dopant-free HTMs generally exhibit enhanced humid and thermal stability [69,70,71,72]. Comparing the face-on and edge-on orientations, the former one is more conducive to the hole extraction and transport as discussed above. Therefore, minimized charge accumulation at perovskite/HTL can be obtained [73]. On the one hand, reduced charge accumulation is beneficial for suppressing charge carrier recombination and thereby higher *V*_OC_ [74]. On the other hand, while the underlying mechanism is complicated, the charge accumulation has been related to the photo-induced degradation of PSCs [75]. Byeon et al. analyzed the degradation process of PSCs working under light illumination, and it was identified that the degradation started from the interface with accumulated charges [76]. DuBose et al. suggested that the localized holes at the iodide sites would induce the formation of I^−^, resulting in halide segregation [77]. In addition, enhanced ion migration was observed for HTMs with lower hole mobility [78]. Recently, based on the result of theoretical calculations, Tong et al. proposed that both the diffusion barrier and the migration length for I^−^ migration could be reduced by the injected holes [79]. In this respect, the face-on orientation is more desirable for the operational stability of PSCs. For example, Cheng et al. reported three HTMs with the ratio of face-on varied in the trend of BDT-DPA-F > BDT-TPA-F > BDT-TPA. After a continuous maximum power point (MPP) tracking for 1200 h, the BDT-DPA-F-based PSC maintained 82.6% of its initial PCE, while the BDT-TPA-F and BDT-TPA-based PSCs showed PCE retentions of 74.3% and 69.4%, respectively [31], which is consistent with the trend of the ratio of face-on. Enhanced operational stability was also observed for SMe-TPA-CuPc-based PSCs when molecular orientation was transformed from edge-on to face-on [33].

Mechanical stability is another issue needed to be addressed for long-life PSCs, especially for the flexible ones [80,81]. It has been demonstrated that the perovskite/HTL interface is the most mechanically vulnerable part in the PSC, and strengthened interface adhesion is desirable [82,83]. In comparison to the edge-on orientation, the face-on orientation is beneficial for a larger contact area between HTM molecules and perovskite, and thereby, a stronger interface interaction can be realized [84]. In 2021, Javaid et al. calculated the adsorption energies of metal phthalocyanines on a MAPbI_3_ surface. It was revealed that the adsorption energy could be up to about 2.6 eV in a face-on adsorption case, while it could be below 0.4 eV in the edge-on case [85]. In 2017, Kim et al. obtained a tetra-tert-butyl substituted copper (II) phthalocyanine HTL with face-on orientation via engineering the interface structure. In the tape test, as shown in Figure 4, both spiro-OMeTAD and PTAA were removed by the adhesive tape. However, only the Au electrode was detached for the phthalocyanine based PSC, which demonstrated the stronger adhesion of phthalocyanine HTL to perovskite surface [86].

## 3. Controlling Strategies

Based on the analyses in Section 2, molecular orientation in organic dopant-free HTMs plays a key role in determining both device performance and the stability of PSCs. Unsatisfactorily, dopant-free HTMs frequently adopt the unsuitable edge-on molecular orientation [87,88,89,90]. Copper (II) phthalocyanine (CuPc), for instance, is a p-type organic semiconductor with high intrinsic hole mobility [91]. However, CuPc adopts an edge-on orientation on the perovskite substrate, resulting in inferior device performance [92]. In this sense, a deeper understanding of factors influencing molecular orientation could provide guidelines to reach the full potential of dopant-free HTMs.

### 3.1. Molecular Structure

The low intrinsic hole mobility of spiro-OMeTAD mainly originates from the bulky spirobifluorene core with 3D configuration. Accordingly, replacing the spirobifluorene core with coplanar ones is a straightforward and effective strategy to improve the intrinsic conductivity of HTMs. Azmi et al. synthesized three new HTMs containing the planar di(1-benzothieno)[3,2-b:2′,3′-d]pyrrole (DBTP) moiety as core unit, while methoxy-substituted diphenylamine (DPA) in the spiro-OMeTAD was retained [93]. The two-dimensional grazing-incidence X-ray diffraction (2D GIXRD) patterns indicated that DBTP-based HTMs adopted the face-on orientation in the film state. In contrast, spiro-OMeTAD showed no preferential molecular orientation. Furthermore, DBTP based HTMs achieved comparable hole mobilities (>10^−4^ cm^2^ V^−1^ s^−1^) to doped spiro-OMeTAD.

Phthalocyanine (Pc) derivatives are a type of small molecule composed of a coplanar conjugated macrocycle ring with 18-π electrons. Owing to the strong coplanarity of the molecular conformation and the richness in π electrons, phthalocyanine molecules can form intimate π-π stacking in solids form, endowing them with high intrinsic charge carrier mobilities [94]. Moreover, Pcs are well known for their excellent thermal stabilities [95]. Hence, phthalocyanines such as CuPc are good choices for dopant-free HTMs in PSCs. However, the pristine CuPc tends to adopt edge-on orientation on the perovskite surface. In this regard, Yang et al. introduced eight methyl groups to the phthalocyanine ring, obtaining CuMe_2_Pc (Figure 5a) [92]. After being thermal evaporated onto the perovskite layer, the GIXRD patterns demonstrated that CuPc and CuMe_2_Pc adopted edge-on and face-on orientations in the film, respectively. Due to the face-on orientation, the CuMe_2_Pc film exhibited a nearly two order of magnitude higher hole mobility than CuPc film (4.79 × 10^−2^ vs. 7.25 × 10^−4^ cm^2^ V^−1^ s^−1^). Thereby, enhanced short-circuit current density (*J*_SC_) and FF were obtained for CuMe_2_Pc based device (Figure 5b), leading to much higher PCE (15.73% vs. 12.55%). In addition, the face-on orientation of the CuMe_2_Pc film resulted in a more condense morphology with a contact angle of 119.6°, which is larger than that of the CuPc film (81.2°). As a result, the unencapsulated CuMe_2_Pc device retained 95% of initial PCE after being stored under conditions with temperature of 25 °C and relative humidity (RH) of 50% for 2000 h. Under the same conditions, the PCE retention for the CuPc device was 76%. The face-on orientation was retained upon varying the central metal to Pd, Zn or metal free [96,97]. In contrast, the orientation turns to edge-on for the tetra-methyl (CuMePc)- and tetra-ethyl (CuEtPc)-substituted CuPc [98], suggesting that the substitution group plays a more significant role than the central metal.

Pristine CuPc has poor solubility in common organic solvents due to the strong intermolecular π-π interactions. To obtain solution-processable phthalocyanine HTMs, substitution groups with sufficient steric hindrance are introduced to the phthalocyanine ring to weaken the π-π interactions. However, weakened intermolecular interactions do not guarantee a face-on molecular orientation. For example, tetra-tert-butyl substituted CuPc is a commercialized Pc derivative with reasonable solubility in organic solvent, but it forms edge-on stacking on the perovskite layer [99]. In 2019, Hu and co-workers used mono-n-butyl-substituted zinc(II) phthalocyanine (H_6_Bu-ZnPc) and hexamethyl-mono-n-butyl-substituted ZnPc (Me_6_Bu-ZnPc) as HTMs for PSCs [100]. Both H_6_Bu-ZnPc and Me_6_Bu-ZnPc are soluble in organic solvent, and the latter can form stronger π-π stacking with face-on orientation. Moreover, the perovskite layer stacked with Me_6_Bu-ZnPc HTL showed a blue-shifted photoluminescence (PL) peak due to decreased spontaneous nonradiative recombination from trap states. This indicates that there is a certain interaction between Me_6_Bu-ZnPc and the perovskite layer, which passivates the defects on the perovskite surface. The same group also explored the effects of alkyl length in soluble tetra-alkyl substitued Cu(II) phthalocyanines [101,102]. It was found that CuEtPc (tetra-ethyl substitued), CuBuPc (tetra-butyl substitued) and CuHePc (tetra-hexyl substitued) adopted edge-on orientation on the perovskite layer but with different title angles [101]. As shown in Figure 6a, the title angle decreased from 46.56° for CuEtPc to 39.58° and 34.00° for CuBuPc and CuHePc, respectively. Furthermore, the length of alkyl chains showed influences on the π-π stacking of phthalocyanine molecules. According to selected-area electron diffraction (SAED) measurements, the strongest π-π stacking with a distance of 3.43 Å was observed for CuBuPc, and it was increased to 3.52 and 3.67 Å for CuEtPc and CuHePc, respectively. The lowest π-π stacking for CuHePc was ascribed to the steric hindrance of hexyl chains. The hole mobilities of these three HTMs were 2.04 × 10^−3^, 1.26 × 10^−3^ and 8.77 × 10^−4^ cm^2^ V^−1^ s^−1^. However, their PCEs were still lower than the doped spiro-OMeTAD mainly due to the lower *V*_OC_ (Figure 6b). It is noted that CuEtPc, CuBuPc, and CuHePc exhibited more upward shifted HOMO levels in the solid state than those in the solutions. For example, the HOMO level of CuBuPc was −5.14 eV in solution, while it was −4.84 eV for solids. For tetra-propyl-substituted phthalocyanine (CuPrPc) [102], simultaneous existence of face-on and edge-on orientation was observed in the film. The hole mobility of CuPrPc HTL was measured to be 2.16 × 10^−3^ cm^2^ V^−1^ s^−1^. Furthermore, the CuPrPc device achieved slightly higher PCE than the spiro-OMeTAD device (17.80% vs. 17.50%).

In a report of Oh et al. [103], the effect of the alkyl side-chain was also studied on two SM-coded molecules (Figure 7a). The result of the density functional theory (DFT) calculation indicated that the hexyl chains increased the steric hinderance between π units in the SM2 molecules and thereby weakened intermolecular stacking. Due to the modulated steric effect, SM molecules showed different stacking behavior in the film as revealed by the two-dimensional grazing-incidence wide-angle (2D GIWAXS) (Figure 7b,c). The mixed orientations were found in SM2 film with a preferential face-on orientation. Eventually, dopant-free SM2 HTM achieved significantly higher hole mobility (1.02 × 10^−3^ cm^2^ V^−1^ s^−1^) and PCE (20.56%) than the SM1 HTM (4.71 × 10^−4^ cm^2^ V^−1^ s^−1^ and 15.37%) (Figure 7d).

Porphyrins (Pors) are analogues to Pcs in regard to the molecular structure. Meanwhile, Pors are thermally and photochemically stable. Therefore, Por-based HTMs also draw wide attention in the field of PSCs [104]. Lee et al. developed zinc (II) Pors (ZnPor) bearing triphenylamine (TPA) and diphenyl-2-pyridylamine (DPPA) at the periphery sites of the Por ring [105]. Although both TPA (PZn-TPA) and DPPA-modified (PZn-DPPA) Pors exhibited face-on orientation, the DPPA group enhances π-π stacking. This results in a slightly higher hole mobility for PZn-DPPA. Alternatively, enhanced face-on stacking can be realized via fluorination of TPA groups [106].

Fluorination was also adopted for benzothiadiazole (BDT)-based small molecules as shown in Figure 8a [31]. For both BDT-TPA and BDT-TPA-F, the large steric hindrance of TPA groups induced significant dihedral angles between the end group and the main chain. However, the electrostatic potential (ESP) analysis revealed that the charge delocalization in the BDT-TPA-F was enhanced due to the strong electronegative of F atoms. Meanwhile, fluorination enhanced the intermolecular C-H···S and C-H···N interactions and introduced extra intermolecular interactions, including C-H···F hydron bonds. The intermolecular interactions was further enhanced by replacing the TPA groups with more coplanar DPA groups. The 2D GIWAXS measurement suggested that the stronger π-π stacking and higher ratio of face-on orientation were obtained for BDT-DPA-F. Upon the above step-wise optimization in the molecular structure, the hole mobility was increased from 5.51 × 10^−5^ cm^2^ V^−1^ s^−1^ for BDT-TPA to 8.40 × 10^−4^ cm^2^ V^−1^ s^−1^ (BDT-TPA-F) and 2.05 × 10^−3^ cm^2^ V^−1^ s^−1^ (BDT-DPA-F). The hole mobility of BDT-DPA-F is comparable to the doped spiro-OMeTAD (3.43 × 10^−3^ cm^2^ V^−1^ s^−1^) in efficient PSCs. Encouragingly, the dopant-free BDT-TPA-F-based device exhibited a PCE of 23.12% (Figure 8b) and T_80_ lifetimes over 1200 h under operational or thermal aging at 85 °C (Figure 8c,d). In comparison, relatively lower PCE retention was observed for BDT-TPA (69.4%) and BDT-TPA-F (74.3%) PCEs under operational aging. The effect of the number of fluorine atoms was explored by Yun et al. [107]. Using a donor (D)-acceptor (A) structured molecule DTS(BTTh_2_)_2_ as the template, the number of fluorine atoms on the A unit varied from 0 (0F) to 1 (1F), 2 (2F), 3 (3F) and 4 (4F). Upon fluorination, consistent with the results of BDT-based HTMs, the hole mobilities of DTS(BTTh_2_)_2_-based molecules were significantly increased owing to the enhanced crystallinity. Especially, 2F and 4F displayed maximum mobilities of 0.22 and 0.06 cm^2^ V^−1^ s^−1^, respectively. However, fluorination did not transfer the edge-on orientation to the face-on orientation.

Introduction of intramolecular non-covalent interaction is an effective way to improve the coplanarity of the backbone and thereby the intermolecular π-π stacking without extra synthetic ring-fusion steps. Based on the above concept, Yang et al. synthesized a molecule coded as BTORCNA, in which an intramolecular S···O interaction can be formed between adjacent thiophene units [108]. In the film state, BTORCNA exhibited improved crystallinity and preferred face-on orientation with a tilt angle of about 35°. Finally, BTORCNA achieved a PCE of 20.59% in p-i-n PSCs. Moreover, the synthetic cost of BTORCNA is about USD 58 g^−1^, much lower than that of spiro-OMeTAD and PTAA.

In D-π-bridge-A type HTMs, the energy level can be tuned easily by modulating the electron donating/accepting ability of D/A. In addition, strong dipolar interactions can be formed between molecules, making them promising HTMs with high mobility. Based on the planar triazatruxene donor and malononitrile acceptor, Rakstys et al. developed three D-π-A HTMs with different π-bridges (Figure 9a) [109]. It was shown that only KR321 exhibited face-on stacking (Figure 9b) with the highest hole mobility of 2.6 × 10^−4^ cm^2^ V^−1^ s^−1^. The PSC based on KR321 achieved a PCE of 19%, which was identical to the doped spiro-OMeTAD. Upon replacing the malononitrile acceptor with 1,3-indandione, N-ethyl rhodanine, and dicyanovinylene N-ethyl rhodamine (Figure 9a) [61], HTMs coded as CI-B1, CI-B2, and CI-B3 were obtained, respectively. Both CI-B1 (Figure 9c) and CI-B2 (Figure 9d) formed randomly arranged stacking in the film, and the former adopted mixed molecular orientations while the latter exhibited only edge-on orientation. For CI-B3 (Figure 9e), it formed edge-on stacking with the highest ordering. Although CI-B1 exhibited the lowest conductivity, the largest PL quenching of the perovskite/CI-B1 stack indicated the fastest hole extraction, which was ascribed to the face-on orientation. Finally, PCEs of 14.78%, 15.33%, and 17.54% were achieved for CI-B1, CI-B2, and CI-B3, respectively. A similar HTM coded as CI-TTIN-2F was also developed (Figure 9a), in which the 2-(5,6-difluoro-3-oxo-2,3-dihydro-1H-inden-1-ylidene)malononitrile (IN-2F) acceptor was used to replace the malononitrile acceptor in KR321 [110]. The intention of choosing IN-2F as the acceptor was based on the fact that it can passivate the under-coordinated Pb^2+^ ions on the perovskite surface via multimodal Lewis acid–base interactions. In the film, CI-TTIN-2F molecules adopted both edge-on and face-on orientations with a hole mobility of 3.7 × 10^−4^ cm^2^ V^−1^ s^−1^.

Based on the above discussion, it can be noted that molecular orientation can be modulated via the tunning of intermolecular interactions. However, the actual effectiveness depends on the core of HTMs and the degree of enhancement/reduction in the interaction strength. For example, the increased steric hindrance in SM2 resulted in the face-on orientation. In contrast, reduced steric hindrance in BDT-DPA-F is conducive for increasing the ratio of face-on orientation. Introducing eight methyl groups to the ring of CuPc resulted in the transformation of the molecular orientation from edge-on to face-on, but the tetra-methyl substituted CuPc still adopted edge-on orientation. In this sense, the preferred molecular orientation cannot be explained by molecular structure alone. 

### 3.2. External Factors

Molecular orientation could also be influenced by properties of the substrate. For example, different molecular orientations were observed for Pcs when the substrate was changed from F-doped SnO_2_ (FTO) to perovskite [92,96]. Shibayama et al. found that spiro-OMeTAD molecules formed random orientation on Si substrate, but showed ordered orientation on NiO_x_, FTO, and perovskite substrates [111]. The result was confirmed by both the surface-insensitive GIWAXS and surface-sensitive near-edge X-ray absorption fine structure (NEXAFS) measurements, indicating that molecular orientation is determined by the substrate properties rather than by the air interface. Research on liquid crystalline metal phthalocyanines has revealed that molecular orientation is controlled by both the substrate/molecule interface and the molecule/air interface, and the latter one is the stronger modulator for films with μm-scaled thickness. However, for the film with a thickness of several tens of nanometers, the substrate is the stronger modulator [112]. In 2021, Stecker et al. investigated the adsorption of CuPc on a non-stoichiometric MAPbI_3_ perovskite surface via scanning tunneling microscopy (STM) [113]. It was found that a single, isolated CuPc molecule adopted face-on orientation on the MAPbI_3_ surface, which is consistent with the DFT calculation result. However, molecules were edge-on oriented in CuPc aggregates. The authors proposed that the above results were due to the fact that the CuPc-MAPbI_3_ interaction is weaker than the intermolecular interactions.

In 2021, Zhou et al. exploited 2,9-diphenyldinaphtho[2,3-b:2′,3′-f]thieno[3,2-b]thiophene (DPh-DNTT) as HTM in p-i-n PSCs, and the effect of substrate temperature during thermal evaporation was studied [54]. As shown in Figure 10a, the obtained film contained a higher face-on ratio and increased vertical hole mobility upon reducing the temperature from 160 °C (HT-DPh-DNTT) to room temperature (RT-DPh-DNTT). In addition, the change in molecular orientation induced a downward shift in HOMO level from −5.03 eV for the HT-DPh-DNTT film to −5.42 eV for RT-DPh-DNTT (Figure 10b), which resulted in enlarged *V*_OC_ (Figure 10c). Meanwhile, the hole mobility increased upon the reduction in substrate temperature, and the RT-DPh-DNTT film obtained the highest hole mobility of 3.3 × 10^−2^ cm^2^ V^−1^ s^−1^. Finally, the device based on the RT-DPh-DNTT film achieved the best PCE of 20.18%. Post thermal treatment may induce the transformation of molecular orientation as reported by Qu et al. [33]. In this work, a Pc molecule (SMe-TPA-CuPc) was used as HTM and showed an almost edge-on orientation in the as-formed film. After being treated at 85 °C for 22 h, molecular orientation was transferred to face-on, leading to improved hole mobility and energy level alignment. In combination with the introduction of an additive into the perovskite film, a PCE of 23% was achieved based on the annealed HTM. In addition, the optimized device retained 96% of the initial PCE after being stored in the dark at 85 °C for 3600 h, while the PTAA device lost 10% PCE after 1000 h. 

Based on the above reports, it can be seen that external factors play a significant role in determining molecular orientation, but related reports are few. According to the research in organic solar cells, processing methods and solution composition may also influence molecular orientation [46]. Undoubtedly, more efforts are deserved in exploring the effect of external factors.

## 4. Conclusions and Outlook

This review summarizes the influences and impacting factors of molecular orientation in small molecular dopant-free HTMs. Compared to the edge-on orientation, face-on orientation is more beneficial for hole extraction, hole transport, and interface energy level alignment. Meanwhile, both enhanced photo stability and mechanical stability can be expected for face-on orientation compared to edge-on orientation. Therefore, face-on orientation is more desirable for efficient and stable PSCs. Molecular orientation in HTL is the result of multi-factorial conjunction. While further research ise deserved to reveal the underlying mechanisms of synergism, the significant role of molecular structure, thermal treatment, and substrate properties have been demonstrated.

From the viewpoint of molecular design, the relative strength between molecule–molecule interactions and molecule–substrate interactions should be taken into consideration when the face-on orientation is pursued. Generally, the relatively stronger molecule–substrate interaction versus the molecule–molecule interaction is beneficial for face-on orientation. On the one hand, intermolecular interactions can be weakened by increasing the intermolecular steric hindrance. On the other hand, molecule–substrate interactions can be enhanced by introducing atoms (F, S, O, Se, et al.) and/or groups (malononitrile, 1,3-indandione, pyridine, et al.) with a passivation effect to form extra interactions with perovskite. Although it was demonstrated to be effective in several cases, molecular engineering should be conducted under deliberative design. This is because both molecule–molecule interactions and molecule–substrate interactions can be simultaneously changed upon variation in the molecular structure. For example, a too large steric hindrance is unfavorable for both molecule–molecule interactions and molecule–substrate interactions. As a result, the pursued molecular orientation may not be obtained. Moreover, the true economic competitiveness of a new HTM depends both on its efficiency and synthetic cost. Hence, synthetic steps and purification procedures of the targeted molecule should be simplified as much as possible.

Alternatively, face-on orientation can be obtained via interface engineering according to the factor that molecular orientation is sensitive to the surface properties of substrates. Substrate modification has been demonstrated to be an effective strategy to modulate molecular orientation in the field of organic solar cells [46]. Furthermore, the surface of perovskite contains numerous defects that could be harmful to the efficiency and stability of PSCs [114]. Therefore, surface passivation is necessary for efficient and stable PSCs, and interface engineering is a common method to address this issue [115]. More importantly, the dosage of interface materials is extremely low [116]. Although the development of HTMs with passivation effect is also an option that can realize the control of molecular orientation and surface passivation simultaneously, the synthetic cost will increase inevitably due to the complicated molecular design and synthetic steps. Hence, we think that interface engineering is a cost-effective strategy that can control the molecular orientation of small molecular HTMs.

Considering the fact that most efficient dopant-free HTMs contain large π-conjugations, it is straightforward to introduce π-conjugated interfacial materials as templates that can induce face-on orientation via strong π-π interactions with HTMs. For example, the templating effects of graphene materials have been demonstrated for different molecules, such as pentacene, DNTT, and copper phthalocyanine [117,118,119]. In addition, graphene materials are widely used interfacial materials in PSCs, and their functions including defect passivation and charge transport optimization have been well demonstrated [120]. In this regard, graphene material is an ideal candidate for molecular orientation control.

## Figures and Tables

**Figure 1 molecules-28-03076-f001:**
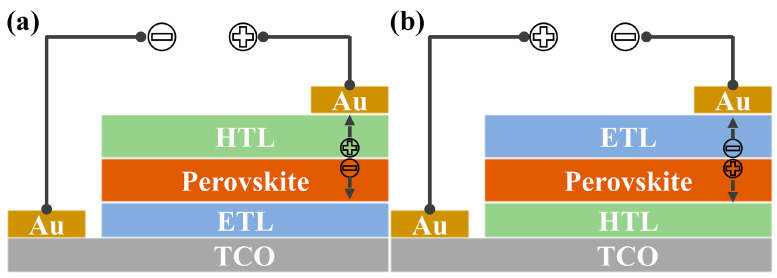
Device configurations of n-i-p (**a**) and p-i-n (**b**) PSCs.

**Figure 3 molecules-28-03076-f003:**
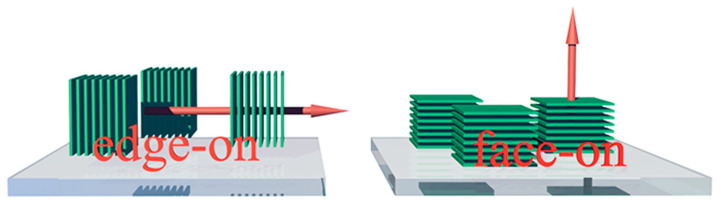
The configuration of molecules in edge-on (**left**) and face-on (**right**) stackings. The red arrow indicates the preferred charge carrier transport direction.

**Figure 4 molecules-28-03076-f004:**
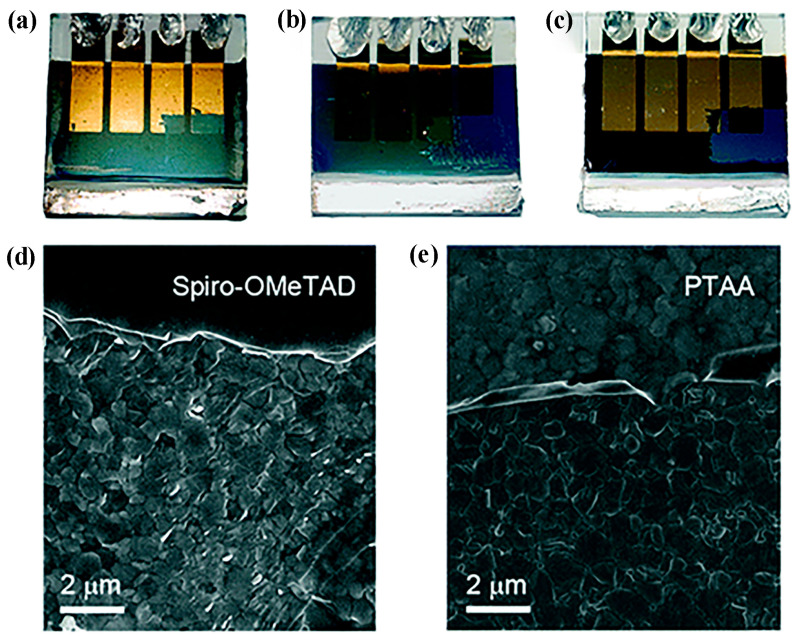
Images of cells after tape test. (**a**) Phthalocyanine-based device; only Au electrode was detached. (**b**) Spiro-OMeTAD-based device; spiro-OMeTAD HTL was partially detached. (**c**) PTAA-based device; PTAA HTL was totally detached. Scanning electron microscope (SEM) images of damaged regions of spiro-OMeTAD- (**d**) and PTAA- (**e**) based devices. Adapted with permission from Ref. [86]. Copyright 2017 The Royal Society of Chemistry.

**Figure 5 molecules-28-03076-f005:**
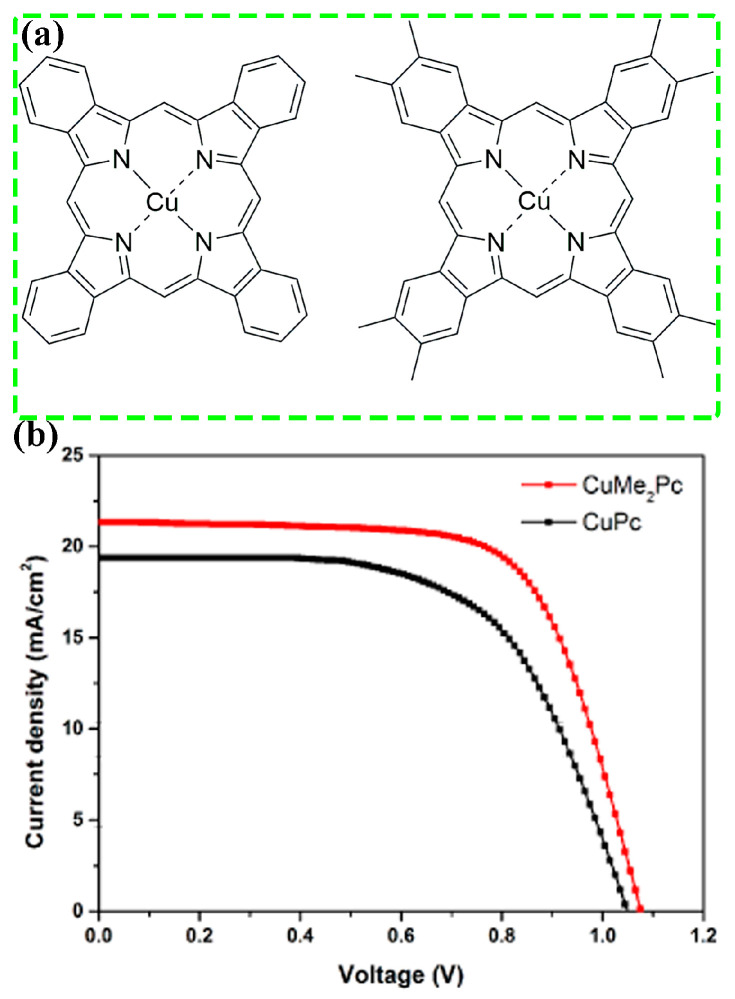
(**a**) Molecular structures of CuPc and CuMe_2_Pc. (**b**) *J*-*V* curves of PSCs based on CuPc and CuMe_2_Pc HTMs. Adapted with permission from Ref. [92]. Copyright 2017 Elsevier.

**Figure 6 molecules-28-03076-f006:**
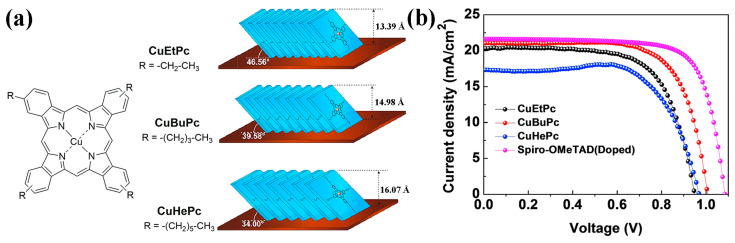
(**a**) Molecular orientation models for Pcs with different alkyl chains deposited on FTO/SnO_2_/perovskite. (**b**) *J*-*V* curves of PSCs with different HTMs. Adapted with permission from Ref. [101]. Copyright 2019 Elsevier.

**Figure 7 molecules-28-03076-f007:**
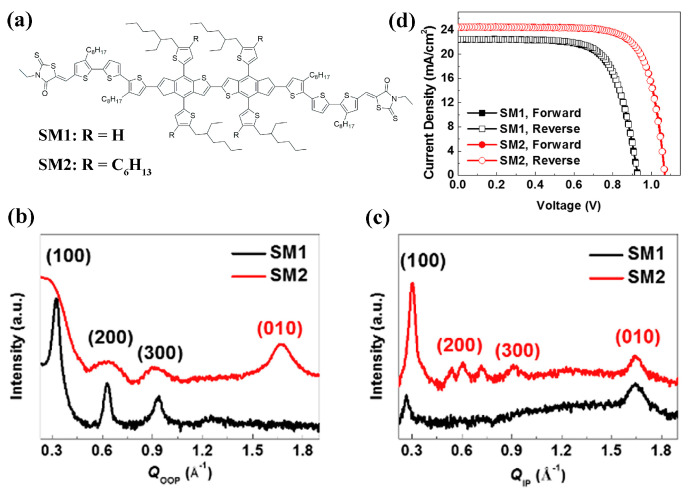
(**a**) Molecular structures of SM1 and SM2. The line-cut profile of HTMs from 2D GIWAXS patterns in the (**b**) out-of-plane (OOP) and (**c**) in-plane (IP) direction. (**d**) *J*-*V* curves of SM-based devices measured under forward and reverse voltage scans (forward direction: from 0 V to forward bias, reverse direction: from forward bias to 0 V). Adapted with permission from Ref. [103]. Copyright 2020 Elsevier.

**Figure 8 molecules-28-03076-f008:**
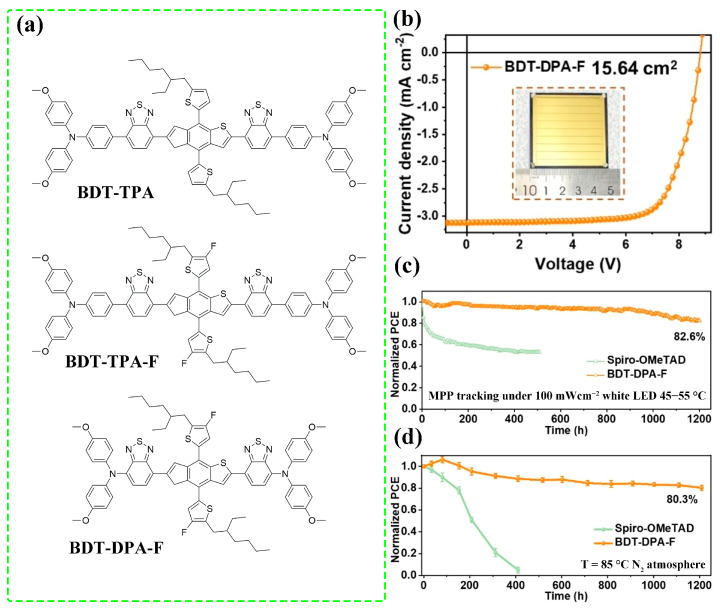
(**a**) Molecular structures of BDT-TPA, BDT-TPA-F, and BDT-DPA-F. (**b**) *J*-*V* curves of the champion devices with different HTMs. (**c**) Operational and (**d**) thermal stabilities of the Spiro-OMeTAD and BDT-DPA-F-based devices under continuous illumination at maximum power point (MPP) tracking conditions and at 85 °C, respectively. Adapted with permission from Ref. [31]. Copyright 2022 John Wiley and Sons.

**Figure 9 molecules-28-03076-f009:**
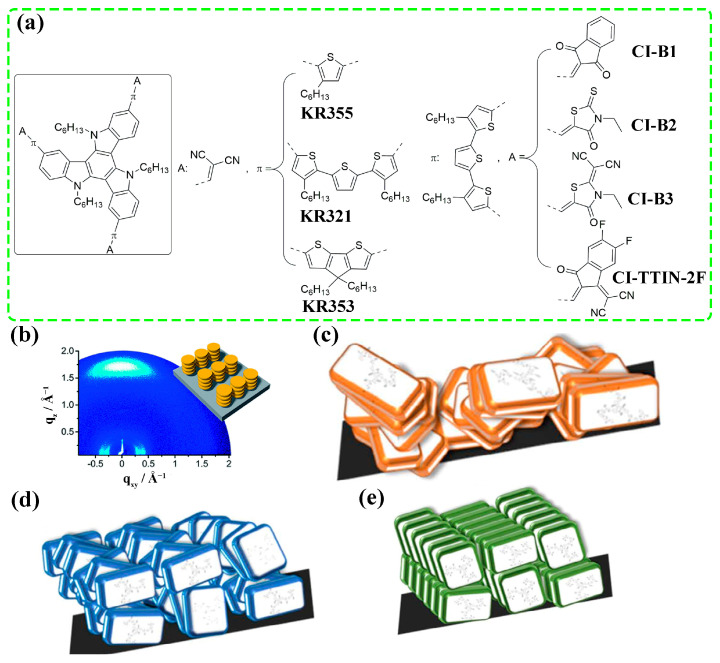
(**a**) Molecular structures of HTMs based on triazatruxene core. (**b**) GIWAXS pattern and surface stacking of the KR321 film on a silica wafer. Adapted with permission from Ref. [109]. Copyright 2017 The Royal Society of Chemistry. Surface stacking of CI-B1 (**c**), CI-B2 (**d**), and CI-B3 (**e**) on silica wafer. Adapted with permission from Ref. [61]. Copyright 2020 John Wiley and Sons.

**Figure 10 molecules-28-03076-f010:**
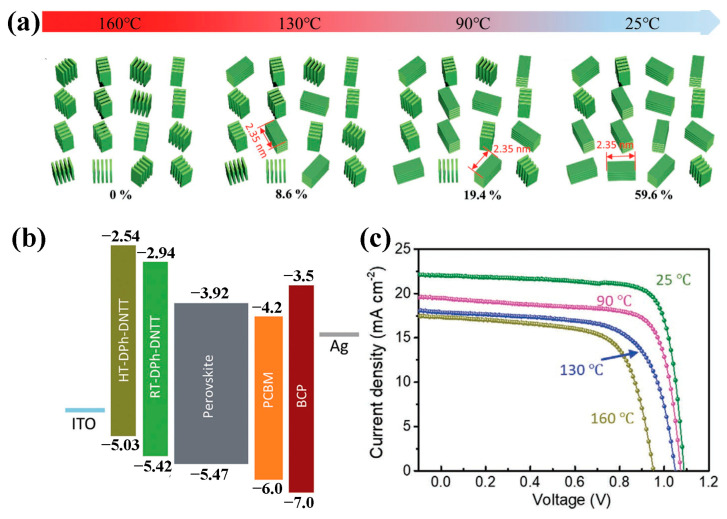
(**a**) The dependence of face-on ratio on the temperature of substrate during evaporation. (**b**) Energy level alignments in p-i-n devices with different HTLs. (**c**) *J*-*V* curves of devices based on different HTLs. Adapted with permission from Ref. [54]. Copyright 2021 John Wiley and Sons.

## Data Availability

No new data were created or analyzed in this study. Data sharing is not applicable to this article.

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
