# Peer review of "Controlling Molecular Orientation of Small Molecular Dopant-Free Hole-Transport Materials: Toward Efficient and Stable Perovskite Solar Cells"

_molecules, 2023, doi:10.3390/molecules28073076_

Round 1

Reviewer 1 Report

This review summarizes some latest progresses on small molecular dopant-free hole-transport materials for perovskite solar cells. The effects of molecular orientation on device efficiency and stability are discussed. This topic is attractive because dopant-free is always pursued to reduce process steps and improve repeatability. However, many parts are lacking of depth and unique insights. This review can be considerable after a major revision. Here are some suggestions.

1)  The “Influences and impacting factors” in title is unclear. A brief and accurate title is very important.

2)  Authors claim “first” outlining analyses on molecular orientation. It is not acceptable because this molecular orientation has been discussed in many similar reviews such as Energy Environ. Sci., 2022, 15, 3630. Energy Environ. Sci. 2020, 13, 4057-4086.

3)  Authors emphasize the main contribution on external factors including thermal treatment and substrates, but they are really short because only one example is summarized. Undoubtedly, more examples and deep discussion are required.

4)  Why is there no discussion on influences of molecular orientation on device stability in the second section?

Author Response

We thank the reviewer for the comments which make our manuscript stronger. The followings are our point-to-point responses to the comments

1.  The “Influences and impacting factors” in title is unclear. A brief and accurate title is very important.

Response:Thanks for this suggestion. We have replaced the title with a new one.

Controlling molecular orientation of small molecular dopant-free hole-transport materials: Toward efficient and stable perovskite solar cells

2. Authors claim “first” outlining analyses on molecular orientation. It is not acceptable because this molecular orientation has been discussed in many similar reviews such as Energy Environ. Sci., 2022, 15, 3630. Energy Environ. Sci. 2020, 13, 4057-4086.

Response: Thanks. We have revised the expression.

This review firstly provides outlining analyses about the important roles of molecular orientation

3. Authors emphasize the main contribution on external factors including thermal treatment and substrates, but they are really short because only one example is summarized. Undoubtedly, more examples and deep discussion are required.

Response: Thanks for this suggestion. Up to now, studies concerning the control of molecular orientation are mainly focused on molecular structure, and there are few reports studied the effect of external factors. We have expanded the discussion on the now available reports.

 4.  Why is there no discussion on influences of molecular orientation on device stability in the second section?

Response: We have added the discussion on photo stability in combination mechanical stability to the revised manuscript.

“In this respect, the face-on orientation is more desirable for operational stability of PSCs. For example, Cheng et al. reported three HTMs with the ratio of face-on varied in the trend of BDT-DPA-F > BDT-TPA-F > BDT-TPA. After a continuous maximum power point (MPP) tracking for 1200 h, the BDT-DPA-F based PSC maintained 82.6% of its initial PCE, while the BDT-TPA-F and BDT-TPA based PSCs showed PCE retentions of 74.3% and 69.4 %, respectively [31], which is consistent with the trend of the ratio of face-on. The enhanced operational stability was also observed for SMe-TPA-CuPc based PSCs when the molecular orientation was transformed from edge-on to face-on [33].

Mechanical stability is another issue needed to be addressed for the long-life PSCs, especially for the flexible ones [75,76]. It has been demonstrated that the perovskite/HTL interface is the most mechanically vulnerable part in the PSC, and strengthened interface adhesion is desirable [77,78]. In comparison to the edge-on orientation, the face-on orientation is beneficial for larger contact area between HTM molecules and perovskite, and thereby stronger interface interaction can be realized [79]. In 2021, Javaid et al. calculated the adsorption energies of metal phthalocyanines on MAPbI3 surface. It was revealed that the adsorption energy could be up to about 2.6 eV in face-on adsorption case, while it was below 0.4 eV in the edge-on case [80]. In 2017, Kim et al. obtained a tetra-tert-butyl substituted copper (II) phthalocyanine HTL with face-on orientation via engineering the interface structure. In the tape test, as shown in Figure 4, both spiro-OMeTAD and PTAA were removed by the adhesive tape. However, only Au electrode was detached for the phthalocyanine based PSC, which demonstrated the stronger adhesion of phthalocyanine HTL to perovskite surface [81].

Reviewer 2 Report

The development of perovskite solar cell is a critical challenge for the green development of modern society. For this target, there is a strong scientific effort for the realization of stable and efficient fourth generation solar cell based on perovskites.

On this purpose, the design of transport material (charge carrier layers) realized in direct contact with the active layer is fundamental for the engineering of commercial systems. The examined manuscript provides a description of the state of art about hole-transport layers based on small molecular dopant free materials. Different systems are examined, and the control of the orientation is considered fundamental in the manuscript. The authors also illustrate the techniques for the realization and the orientation control of hole-transport layers.

The manuscript can be published in the journal Molecules.

Minor issues:

11.  CTL acronym, reported at line 36, is defined at line 40. Please move the acronym definition.

22.    Figure 6d forward and reverse reported in the legend is not defined in the caption.

33.   Line 373 and 381, Authors probably refer to Figure 8a and not figure 9a.

Author Response

We thank the reviewer for the positive comment on our manuscript. The followings are point-to-point responses to the comments.

1. CTL acronym, reported at line 36, is defined at line 40. Please move the acronym definition.

Response: Thanks. We have moved the CTL acronym definition to the correct position.

2. Figure 6d “forward” and “reverse” reported in the legend is not defined in the caption.

Response: Thanks. We have added the definition of “forward” and “reverse” to the caption.

J-V curves of SM based devices measured under forward and reverse voltage scans (forward direction: from 0V to forward bias, reverse direction: from forward bias to 0V)

3. Line 373 and 381, Authors probably refer to Figure 8a and not figure 9a.

Response: Thanks. We have revised this mistake.

Reviewer 3 Report

This is an interesting paper, that looks at one aspect that is not discussed a lot in perovskite research: molecular orientation!

Comments

1

A paragraph on the theoretical background of the reason why molecular orientation is important is missing. Inclusion of this would make the paper stronger.

We can suggest the following papers that described this for interactions between chromophoric groups.  They highlight the effects of the orbitals, and charge transfer integrals. See for instance:

https://doi.org/10.1039/c0pp00050g

https://doi.org/10.3390/molecules27030891

2

The approach on page 5 is valid, but in a way simplistic. Next to face-on and edge-on, there is also an effect of tilt angle, twist angle, and slip-stack configurations.

Please indicate a bit more on various aspects.

https://doi.org/10.1021/ja508814z

https://doi.org/10.1021/acs.accounts.0c00590

3

Please make the Future Outlook more specific, suggesting some good strategies for specific molecule-surface combinations that you deem most successful (like graphite-pyrene).

4

Figure 5 is too small.

5

Somehow the headings and subheading do not make things very clear: See:

2. Influences of molecular orientation

2.3. Influences of molecular orientation on device stability

3. Impacting factors influence molecular orientation

3.1. Influences of molecular orientation on device stability

Please try to improve this, by making more clear what the sections are about.

Author Response

We thank the reviewer for the posivite comments on our manuscript. The followings are our poin-to-point responses to the comments.

1. A paragraph on the theoretical background of the reason why molecular orientation is important is missing. Inclusion of this would make the paper stronger.We can suggest the following papers that described this for interactions between chromophoric groups.  They highlight the effects of the orbitals, and charge transfer integrals. See for instance:

https://doi.org/10.1039/c0pp00050g

https://doi.org/10.3390/molecules27030891

Response: Thanks for this suggestion. We have added a description of theoretical background to explain the importance of molecular orientation in determining the interfacial charge transfer.

Similar to the intramolecular charge transport between the electron donor part and the electron acceptor part [55], the interfacial charge transfer depends on the effective electronic coupling, Veff, at the interface [56]. For the charge transfer between diabatic states i and f, the relationship between transfer rate ki→f and Veff is described as , where  is the density of states [56]. Upon depositing a molecule onto a substrate, the interfacial Veff depends on the molecular orientation.

2. The approach on page 5 is valid, but in a way simplistic. Next to face-on and edge-on, there is also an effect of tilt angle, twist angle, and slip-stack configurations.

Please indicate a bit more on various aspects.

https://doi.org/10.1021/ja508814z

https://doi.org/10.1021/acs.accounts.0c00590

Response: Thanks for this suggestion. Factors such as tilt angle, twist angle, and slip-stack configurations are packing details of organic semiconductors, and they determine the rate of charge transport in solids. While the molecular orientation is referred to the stacking orientation of molecules with respect to the substrate, which determines the match between charge transport channels of organic semiconductor and certain devices. We have added discussion on packing details to the revised manuscript.

In this respect, the charge carrier transport in organic solids depends on packing details of molecules which determines the intermolecular π-π overlap. For example, the lamellar packing exhibits larger π-π overlap than the herringbone packing [47]. In lamellar packing motif, the π-π overlap degree varies upon the slip displacement and slip angle which depends on the molecular structure [48,49]. Although the charge carrier transport depends on the packing details of molecules, the most efficient charge carrier transport channel is in the direction parallel to the π-π stacking orientation [50,51].

3. Please make the Future Outlook more specific, suggesting some good strategies for specific molecule-surface combinations that you deem most successful (like graphite-pyrene).

Response: Thanks for this suggestion. After more concrete literature review, we think that graphene materials are ideal interfacial materials which can act as template for face-on orientation. We have added the corresponding discussion to the revised manuscript.

Considering the fact that most efficient dopant-free HTMs contain large π-conjugation, it is straightforward to introduce π-conjugated interfacial materials as template, which can induce face-on orientation via strong π-π interaction with HTMs. For example, the templating effect of graphene materials have been demonstrated for different molecules, such as pentacene, DNTT, and copper phthalocyanine [111-113]. In addition, graphene materials are widely used interfacial materials in PSCs, and their functions including defect passivation and charge transport optimization have been well demonstrated [114]. In this regard, the graphene material is an ideal candidate for molecular orientation control.

4. Figure 5 is too small.

Response: This figure has been replaced with the larger one.

5. Somehow the headings and subheading do not make things very clear: See:

  1. Influences of molecular orientation

2.3. Influences of molecular orientation on device stability

  1. Impacting factors influence molecular orientation

3.1. Influences of molecular orientation on device stability

Please try to improve this, by making more clear what the sections are about.

Response: Thanks for this suggestion. We have replaced headings and subheadings with clearer ones.

Round 2

Reviewer 1 Report

I think the revised manuscript is suitable for the Journal.